# Multiplex Detection of 24 Staphylococcal Enterotoxins in Culture Supernatant Using Liquid Chromatography Coupled to High-Resolution Mass Spectrometry

**DOI:** 10.3390/toxins14040249

**Published:** 2022-03-31

**Authors:** Donatien Lefebvre, Kevin Blanco-Valle, Jacques-Antoine Hennekinne, Stéphanie Simon, François Fenaille, François Becher, Yacine Nia

**Affiliations:** 1Laboratory for Food Safety, ANSES, Université Paris-Est, 91191 Maisons-Alfort, France; donatien.lefebvre@anses.fr (D.L.); k.blancovalle@gmail.com (K.B.-V.); jacques-antoine.hennekinne@anses.fr (J.-A.H.); 2CEA, INRAE, Département Médicaments et Technologies pour la Santé (DMTS), SPI, Université Paris-Saclay, 91191 Gif-sur-Yvette, France; stephanie.simon@cea.fr (S.S.); francois.fenaille@cea.fr (F.F.); francois.becher@cea.fr (F.B.)

**Keywords:** coagulase-positive staphylococci, staphylococcal enterotoxins, mass spectrometry, signature peptides

## Abstract

Staphylococcal food poisoning outbreaks are caused by the ingestion of food contaminated with staphylococcal enterotoxins (SEs). Among the 27 SEs described in the literature to date, only a few can be detected using immuno-enzymatic-based methods that are strongly dependent on the availability of antibodies. Liquid chromatography, coupled to high-resolution mass spectrometry (LC-HRMS), has, therefore, been put forward as a relevant complementary method, but only for the detection of a limited number of enterotoxins. In this work, LC-HRMS was developed for the detection and quantification of 24 SEs. A database of 93 specific signature peptides and LC-HRMS parameters was optimized using sequences from 24 SEs, including their 162 variants. A label-free quantification protocol was established to overcome the absence of calibration standards. The LC-HRMS method showed high performance in terms of specificity, sensitivity, and accuracy when applied to 49 enterotoxin-producing strains. SE concentrations measured depended on both SE type and the coagulase-positive staphylococci (CPS) strain. This study indicates that LC-MS is a relevant alternative and complementary tool to ELISA methods. The advantages of LC-MS clearly lie in both the multiplex analysis of a large number of SEs, and the automated analysis of a high number of samples.

## 1. Introduction

Staphylococcus enterotoxins (SEs) are potent gastrointestinal exotoxins produced by coagulase-positive staphylococci (CPS), including *Staphylococcus aureus* (*S. aureus*). They are resistant to harsh conditions such as heat treatment and low pH that easily destroy CPS, as well as to proteolytic enzymes, and therefore retain their activity in the digestive tract after ingestion [1]. Consumption of food contaminated by a few nanograms of SEs rapidly and abruptly induces symptoms such as nausea, copious vomiting, abdominal cramps, and diarrhea. Reported dose levels of SEs involved in foodborne outbreaks are scarce and variable. The Benchmark Dose lower limit (BMD10) was estimated to be 6.1 ng for SEA type [2]. Since 2011, the annual zoonosis reports of the European Food Safety Authority (EFSA) [3] have highlighted SEs as a serious causative agent of foodborne outbreaks. Thus, SEs are assumed to be a threat to public health.

SEs constitute a family of more than 27 different serological types, with similarities in structure, function, and sequences. However, emetic activity has only been demonstrated for 19 SEs (SEA, SEB, SEC, SED, SEE, SEG, SEH, SEI, SEK, SEL, SEM, SEN, SEO, SEP, SEQ, SER, SES, SET, and SEY) [4,5,6,7,8]. SElJ, SElU, SElV, SElW, SElX, SElZ, SEl26, and SEl27 have not yet been tested for their emetic activity [9,10,11,12]. The genes encoding the different Ses are located on accessory genetic elements, i.e., prophages, plasmids, pathogenicity islands (SaPIs), enterotoxin gene clusters (egc), and the staphylococcal cassette chromosome (SCC). Recently, the use of whole-genome sequencing (WGS) tools showed that one CPS strain could contain between 1 and 11 types of staphylococcal enterotoxin genes (se) that can simultaneously express several SEs. For example, studies carried out on staphylococcal food poisoning outbreaks have demonstrated that one CPS strain is able to produce at least two types of SEs in contaminated food [13,14,15]. Moreover, the divergence of SEs in term of amino-acid sequences is further increased by genetic and amino-acid variations within each type, which are referred to as toxin variants [16]. As an example, 17 variants were reported for the sole SEI type.

Current detection methods for SEs include bio-assays, mass spectrometry (MS), and immuno-enzymatic assays. The immuno-enzymatic approach is the most common. At present, only the so-called classical enterotoxins (SEA, SEB, SEC, SED, and SEE) can be detected using commercially available detection kits. For SEB, due to its potential dual use, more than 17 commercial kits are available [17]. In food matrices, the five classical SEs can be analyzed according to the EN ISO 19020 procedure using two validated commercial kits: VIDAS^®^ SET2 (bioMérieux^®^, Marcy-l’Étoile, France) and RIDASCREEN^®^ SET Total (R-Biopharm^®^, Darmstadt, Germany).

At present, there are no commercially available reagents or kits allowing for the detection of the other SEs with emetic activity (SEG, SEH, SEI, SEK, SEL, SEM, SEN, SEO, SEP, SEQ, SER, SES, SET, and SEY). In fact, very limited studies have been published on the detection of these SEs [18,19,20,21,22,23,24,25,26,27]. In these studies, the authors mainly focused on the development of ELISA methods using in-house-produced anti-SEs antibodies, which limits their widespread availability.

The mass spectrometry (MS) method is a useful alternative to immune-enzymatic methods. This technique provides high specificity and multiplex analysis of toxins [28]. Targeted MS methods can be optimized for more sensitive detection once the sequences of the proteotypic peptides are characterized [29]. Unlike ELISA methods, LC-MS can be implemented without any specific reagents, which means that there is no need for antibodies. 

Only a few studies have been published on multiplex SEs detection by MS approaches. Concerning the 27 described SEs, studies mainly focused on the detection of the five classical SEs (SEA-SEE). Few studies have been published on the detection of the eight other emetic SEs by LC-MS (SEG, SEH, SEI, SEK, SEL, SEM, SEN, and SEQ) [30,31,32,33,34,35,36,37,38]. No published work was found on the optimization of the detection of the other six emetic SEs (SEO, SEP, SER, SES, SET and SEY) by MS. 

However, no studies have been published on the multiplex detection of all emetic SEs. In addition, the diversity of the sequence variants of SEs was not taken into account during method development, which may have an impact on their specificity. Only Lefebvre et al. 2021 used sequences of eight SEs (SEA to SEE, SEG, SEH and SEI) and their variants for the selection of proteotypic peptides [29].

Last but not least, only the SEA, SEB, SEC, SED and SEE standards can be purchased in lyophilized form and then reconstituted in the laboratory, which can induce a significant bias. As a result, the lack of SEs calibration solution for all SEs prevents the implementation of quantitative methods. To overcome this issue, the so-called “Hi3” approach, using a proven, label-free quantification method, can be implemented [39]. It has been described that the concentration of a protein in a complex mixture can be estimated using the mass spectrometry signal of the three most intense peptides obtained from standard solution. This approach has been used in different studies to analyze the cytoplasm of various mutants of *Staphylococcus aureus* and *Bacillus subtilis*, allowing for the absolute quantification of several proteins of the cytosolic proteome [40,41].

In this study, a label and antibody-free LC-MS method based on the bottom-up proteomics approach was optimized for the multiplex quantification of 24 SE types in medium culture, i.e., SEA, SEB, SEC, SED, SEE, SEG, SEH, SEI, SElJ, SEK, SEL, SEM, SEN, SEO, SEP, SEQ, SER, SES, SET, SElU, SElX, SEY, SElZ, and TSST1. A signature peptide database was established for 24 SE types, considering 162 sequence variants. The LC-MS method was tested on 24 SEs produced in medium culture by a panel of 49 representative CPS strains previously characterized by a WGS tool for their content of *se* genes. The LC-MS method was evaluated in terms of specificity, sensitivity, and accuracy. Finally, SE quantification is essential to study ES toxicity and determine the emetic dose as Benchmark Dose (BMD50). In this work, SE quantification was performed by the label-free Hi3 method to overcome the absence of SE standard solutions. The types of SEs and the concentrations secreted by each CPS strain were estimated, showing the ability of each strain to simultaneously produce several types of SEs at different concentration levels, depending on the SE type.

## 2. Results and Discussion

### 2.1. Se Gene Characterization

Reference CPS strains, types of *se* genes, and their variants are presented in Table 1. More detailed information on CPS strains, such as origin, date of isolation, and type of food from which the strain was isolated, are reported in Appendix A.

Among the 49 CPS strains, 42 (86%) were isolated from food samples, most of which were involved in food poisoning outbreaks, indicating their ability to express SEs. They were sequenced and analyzed using the NAuRA [16] genomic tool. The results showed that each CPS strain harbors between two and 11 *se* genes. The 11 *se* genes contain between two and five genetic variants: *sea* (4), *seb* (4), *sed* (4), *see* (3), *seh* (4), *selj* (4), *sep* (3), *seq* (5), *ser* (4), *ses* (2), and *selz* (4). Eight enterotoxin genes contain between six and ten variants: *seg* (6), *sec* (10), *sek* (8), *sem* (9), *sen* (9), *seu* (10), *sey* (6), and *tsst1* (6). According to gene sequencing data, the four *se* enterotoxin genes *sel*, *sei*, *seo* and *selx* contain 11, 12, 13 and 20 variants, respectively. Finally, the *set* type has no variants. The *sev*, *sew*, *sel26* and *sel27* genes were not detected in the strains analyzed in this work. This may be explained by the rarity of these genes in the CPS strains of this study that were isolated from food samples [16]. In total, the 49 CPS strains that were used to prepare naturally SE-contaminated medium samples harbored 24 *se* genes and their 162 variants.

### 2.2. Establishment of a Signature Peptides Database for Each Enterotoxin Type

The brain heart infusion (BHI) broth samples contaminated by SEs were prepared for each CPS strain and were then submitted to the acid precipitation protocol, tryptic digestion, and LC-MS analysis, as described in Figure 1.

SEs peptides released by trypsin in the contaminated BHI samples were analyzed by LC-MS in data-dependent acquisition (DDA) mode (non-targeted analysis) and were processed by the X!Tandem Pipeline 0.4.3 [42] using both genomic sequences obtained by NAuRA genomic software [16] and DDA data. This step is used to determine all peptides and their corresponding retention times (RT). In total, for the 24 SEs and their variants, a preliminary list of 207 peptides was established (Appendix A). Between four and 13 peptides were detected for each SE type: SEA (12), SEB (12), SEC (14), SED (9), SEE (8), SEG (7), SEH (10), SEI (10), SElJ (6), SEK (10), SEL (8), SEM (7), SEN (6), SEO (8), SEP (5), SEQ (13), SER (11), SES (7), SET (9), SElU (4), SElX (9), SEY (8), SElZ (5) and TSST1 (9). These 207 peptides were fragmented with three different collision energies: 15, 18, and 21% CE. The best collision energy was selected on the extracted-ion chromatogram (XIC) of the three most intense fragments for each peptide. Fragment selection was performed using Skyline version 21.1 software (https://skyline.ms/; accessed date 25 June 2021); only the y ions were analyzed. The fragmentation energies and the selected fragments for further quantification are shown in Appendix A. An example of optimized collision energy for SEA is shown in Table 2. These 207 peptides were analyzed in parallel reaction monitoring (PRM) mode for identification of the best responder peptides, as described in Lefebvre et al. 2021 [29]. Among the best peptides, the most intense and non-interfered y-ion fragments were selected for peptide detection and quantification. At least three specific peptides were selected for each type of enterotoxin, considering SEs variants. Therefore, 93 specific signature peptides covering the 24 SEs and their variants were selected and included in a signature peptide database. Enterotoxins (and their number of specific peptides) are: SEA (5), SEB (3), SEC (4), SED (3), SEE (4), SEG (5), SEH (4), SEI (5), SElJ (4), SEK (4), SEL (4), SEM (4), SEN (3), SEO (3), SEP (4), SEQ (3), SER (3), SES (4), SET (3), SElU (4), SElX (4), SEY (4), SElZ (5) and TSST1 (4) (see Appendix A). An example of the optimization and identification of specific peptides for enterotoxin SEA is described in Table 2. Only one common specific peptide (QNTVPLETVK, *m*/*z* = 564.8166) was detected in the four variants. To avoid negative deviation, this result highlights the importance of optimizing several specific peptides for each type of SE, considering sequence variants.

All peptides reported for SEA to SED in the literature were also detected in the list of 93 peptides [28,43,44,45,46]. For enterotoxins SEA, SEB and SED, the authors determined between two and four specific peptides that we confirmed in this study. Muratovic et al. 2015 [43] reported a set of eight peptides for SEA and eight for SEB. These SEB peptides were identified in our set. For SEA, six of the eight peptides were available in our set.

As a result, the established list of 93 specific signature peptides can be considered as a reference database for further work on the detection of SEs using an LC-MS method, and could be used for further applications in food matrices.

### 2.3. Assessment of Qualitative Performance

The PRM method implemented with the peptide database was selected for further evaluation, considering the higher sensitivity of the targeted acquisition mode [47,48]. The sensitivity, specificity, and accuracy of the developped LC-MS method were calculated for each type of SE based on an analysis of the contaminated BHI samples. The qualitative performances obtained for each type of SE are reported in Table 3.

#### 2.3.1. LC-MS Specificity

Specificity describes the ability to obtain a negative result for a sample that does not contain SEs, regardless of the SE type and the analyzed variant (blank sample). In this work, specificity was determined by comparing the signal detected by PRM with the WGS results. Therefore, the absence of a PRM signal was expected when the *se* gene was not detected by the genomic tool. This was calculated as follows:$$\mathrm{Specificity}\;\left(\%\right)=\frac{\mathrm{Number}\;\mathrm{of}\;\mathrm{samples}\;\mathrm{correctly}\;\mathrm{analyzed}\;\mathrm{as}\;\mathrm{negative}\;\mathrm{for}\;\mathrm{type}\;x}{\mathrm{Number}\;\mathrm{of}\;\mathrm{blank}\;\mathrm{BHI}\;\mathrm{samples}\;}\times 100\;$$
where the number of blank BHI samples is the culture samples prepared with a CPS strain negative for the targeted *se* gene. For example, when the SEA type is analyzed, the blank sample must be negative for the *sea* gene even if it contains other types of *se* gene. The calculate specificity was 100%, irrespective of the type of the SE, meaning the absence of positive deviation. This highlighted the absence of an interference between different enterotoxin types. In fact, the obtained results were negative for the targeted enterotoxin despite the presence of other SE types.

#### 2.3.2. LC-MS Sensitivity

Sensitivity describes the ability to detect the correct type of SE for a sample that contains the analyzed SE type. In this work, sensitivity was calculated assuming that all *se* genes detected by the genomic tool expressed the corresponding SEs in BHI supernatant. For example, if the CPS strain added to and incubated in the sample contains the *sea* gene, the enterotoxin SEA must be detected by LC-MS, even if this CPS is able to produce several types of SE. Therefore, sensitivity for the LC-MS method is calculated as follows:$$\mathrm{Sensitivity}\;\left(\%\right)=\frac{\mathrm{Number}\;\mathrm{of}\;\mathrm{SE}\;\mathrm{types}\;\mathrm{correctly}\;\mathrm{detected}}{\mathrm{Number}\;\mathrm{of}\;\mathrm{BHI}\;\mathrm{samples}\;\mathrm{containing}\;\mathrm{the}\;\mathrm{type}\;\mathrm{of}\;se\;\mathrm{targeted}}\times 100$$

The LC-MS method had a sensitivity of 100% for 20 SEs. However, for the SEL, SEN, SEO and SElU types, method sensitivity was 93.8% (15/16), 76.0% (19/25), 88.5% (23/26) and 90.0% (18/20), respectively. For SEN, SEO and SElU, the observed negative deviations could be due to the very low amount of enterotoxins expressed by *egc* operon. In fact, this operon harbors six types of SEs: SEG, SEI, SEM, SEN, SEO, and SElU (Table 4). The sensitivity of the method was 100% for types SEG, SEI and SEM. The measured concentrations were the lowest and were close to the limit of detection (1 ng/mL), Table 5. This observation was reported by Schwendimann et al. 2021 [49] and confirmed the expression of *egc* enterotoxins at the ng/mL level. Finally, for type SEL, only one negative deviation was observed (15/16), which could be due to lack of expression of the *sel* gene by this strain or to the very low amount produced in the BHI. 

It is also possible to evaluate overall sensitivity for the 24 SEs. In fact, in the 49 analyzed BHI samples, 335 enterotoxins were expected to be detected, assuming that all *se* genes expressed their corresponding SEs in the BHI supernatant. The LC-MS method detected 323 SEs among them, providing an overall sensitivity of 96.3%.

The sensitivity values that were obtained in our study could be considered satisfactory, considering the multiplex detection of 24 SE types and the bias that can be introduced by the non-expression of *se* genes.

#### 2.3.3. LC-MS Accuracy

Accuracy combines specificity and sensitivity performances. It describes the ability to obtain a correct result for samples that are negative for SEs and for samples that contain SEs. It was calculated as follows:$$\mathrm{Accuracy}\;\left(\%\right)=\frac{\mathrm{Number}\;\mathrm{of}\;\mathrm{SE}\;\mathrm{types}\;\mathrm{correctly}\;\mathrm{detected}\;\mathrm{or}\;\mathrm{not}\;\mathrm{detected}}{\mathrm{Total}\;\mathrm{number}\;\mathrm{of}\;\mathrm{analyses}}\times 100\;$$
where the number of SE types that were correctly detected or not detected is the sum of the number of correct positive results when SE was expected to be “detected” and the number of correct negative results when the SE was expected to be “not detected”, and total number of analyses corresponds to the sum of the number of strains containing the targeted *se* genes and number of strains negative for the targeted *se* gene (see Table 3).

The LC-MS method developed in this study showed 100% accuracy for 20 SE types. For SEL, SEN, SEO and SElU, LC-MS accuracy was 98.0% (48/49), 87.7% (43/49), 93.9% (46/49), and 95.9% (47/49), respectively. As explained above, *egc* enterotoxin expression is the only identified source of this deviation.

The overall accuracy of the LC-MS method can be calculated for the multiplex detection of 24 SEs. Therefore, 1164 correct analyses were obtained in the 49 BHI samples over the 1176 analyses. Therefore, the overall accuracy of the multiplex LC-MS method was 99.0%.

Qualitative performance criteria highlighted the high specificity of the optimized LC-MS method dedicated to the detection of 24 SEs. Specificity, sensitivity, and accuracy values can be considered satisfactory regarding the large number of targeted enterotoxins. In the literature, these qualitative performance criteria were mainly reported for ELISA methods and for a very limited number of SEs. Nia et al. 2021 [18] calculated performance criteria for the standard EN ISO 19020 for five classical enterotoxins, SEA, SEB, SEC, SED and SEE, in five categories of food matrices. Specificity varied between 90 and 100%, and sensitivity between 87 and 100%, for the two most efficient immuno-enzymatic assays. Féraudet-Tarisse et al. 2021 [27] developed immuno-enzymatic tests for enterotoxin types SEA, SEG, SEH, and SEI in medium and evaluated specificity on only eight CPS strains. The results showed an absence of cross-reactions and were not interpreted in terms of specificity, sensitivity, and accuracy. For LC-MS methods, specificity and sensitivity were only evaluated in the case of three types of SEs (SEA, SEB, and SED). In this work, performance criteria were calculated using data obtained from a large set of 24 SEs and their variants naturally produced in medium. This work is the first comprehensive evaluation of LC-MS performance for the detection of 24 SEs in medium culture.

### 2.4. SEs Quantification in BHI Culture Supernatant

Only five purified SE standards are commercially available (SEA, SEB, SEC, SED, and SEE). To estimate the concentration levels of the 24 SEs produced in the BHI and to overcome the absence of SE standard solutions, an unlabeled quantification method, known as the “Hi3” procedure, was used in accordance with Silva et al. 2006 [39]. In brief, protein concentration can be estimated from the MS signal of the sum of the three most reactive peptides of each protein compared to the best three peptides of a standard at a known concentration [40,41].

In this study, a mixture of five SE standard solutions (SEA to SEE) was spiked in 12 BHI samples to obtain a calibration curve covering 12 concentrations: 4, 10, 20, 40, 100, 200, 400, 1000, 2000, 4000, 10,000, and 20,000 fmol/mL (corresponding to 0.1, 0.25, 0.5, 1, 2.5, 5, 10, 25, 50, 100, 250, and 500 ng/mL at 25 kda, respectively). The three best quantitative peptides of each SE standard were used for quantification. For signal normalization, a set of 28 heavy peptides was added at constant concentration to all samples and used as the external standard.

The calibration curve, established according to the Hi3 procedure, was tested individually for each type of SE for which standard solutions were available, over the 12 concentrations, in duplicate. Then, biases were calculated and are reported in Table 4 and Figure 2.

The R^2^ of each curve is above 0.95 and the CVs of replicates were below 20% overall. Calculated bias can be interpreted as follows:-At very low concentrations (10 and 20 fMol): over the 10 calculated biases, 6 values were >30%.-Between 40 and 20,000 fMol: SEA showed bias between 3.6 and 22.6%, except at 40 fMol with bias at 37.9%. For SEB, bias varied between 0.5 and 28.6%. For SED, bias varied between −7.5 and −37.8%. For SEE, bias varied between 12.2 and 35.9%. Finally, SEC showed a higher bias compared to SEA, SEB, SED and SEE, with values between 28.1 and 36.5%, except at a concentration of 40 fMol, −1.5%. Finally, among the 45 bias values calculated in this range of the standard curve, 10 values varied between 30% and 40%. However, among these 10 bias values, six were obtained only for SEC.

In conclusion, the calibration curve obtained according to the Hi3 procedure can be considered representative of the four SE types: SEA, SEB, SED, and SEE. For SEC, the calibration curve can be used semi-quantitatively due to the considerable bias observed.

The limit of quantification (LOQ) was determined for the five SEs, based on bias below 20%. For each SE, we observed at least two best peptides showing an LOQ at 40 fMol, corresponding to 1 ng/mL.

In the absence of a standard solution for the other 19 SEs studied in this work, it is difficult to extrapolate the observations obtained for the 5 SEs (SEA to SEE) to the 19 SEs. However, we assume that the standard curve obtained according to Hi3 can be implemented to obtain a semi-quantitative SEs concentration in BHI medium.

### 2.5. Application of the LC-MS Method for Multiplex Analysis of 24 Enterotoxins Produced by CPS Strains in Culture Supernatant

SEs were analyzed in the 49 BHI samples prepared for each CPS strain by the LC-MS method developed using the signature peptides database and the Hi3 calibration curve implemented in this study. Measured concentrations and median obtained for each type of SE are reported in Table 5.

The obtained results showed that individual strains were able to produce between 2 and 11 SEs. Measured concentrations varied between LoQ (<1 ng/g) and 21,943.4 ng/g. Among the 335 enterotoxins that were expected to be produced by strains in supernatant, 323 enterotoxins (96%) were detected by LC-MS. Among them, 215 (67%) were quantified and 108 (33%) were detected below the LoQ (<1 ng/g) (Table 5). However, concentration levels depend strongly on the type of SEs. Therefore, four SE groups can be distinguished depending on their concentration expressed by CPS strains:-Very low level: SEG, SEI, SEN, SEM, SEO and SElU, expressed by the genomic cluster *egc*, were detected at very low levels compared to other SEs. Their concentrations were below 1 ng/g except for SEM (median 2.4 ng/g, *n* = 24). Also, SElJ was produced at very low concentrations. -Low level: the six enterotoxins SEL, SEP, SES, SElX, SEY and SElZ were weakly produced with a median between 5.0 ng/g (SEY, *n* = 10) and 11.6 ng/g (SElX, *n* = 42). It should be noted that type SES was analyzed only twice, at 6 and 18 ng/g. In this group, the concentration measured also depended on the strain.-Medium level: SEA, SED, SEE, SEH, SEK, SEQ, SER, SET, and TSST1 were produced at relatively similar concentrations, with a median between 29.6 ng/g (SER, *n* = 9) and 60.5 ng/g (SEA, *n* = 14). SET was very rarely detected and was analyzed in only one sample at 65.6 ng/g. For these nine enterotoxins, concentrations could vary depending on the strain. For example, for 14 strains, SEA concentration varied between 13.7 ng/g and 124.1 ng/g, with median 60.5 ng/g.-High level: very high concentrations were obtained only for SEB and SEC, with a median of 1731.3 (*n* = 17) and 3285.9 ng/g (*n* = 7), respectively. For SEB, concentrations varied between 710.8 and 21,943.1 ng/g, except for strain 337E (20.2 ng/g). For SEC, concentrations varied between 65.2 and 8364.9 ng/g, except for strain 15SBCL1438 (2.2 ng/g).

The total amounts of SE produced by each strain are reported in Table 5. The lowest value was 3.4 ng/g and the highest 22,103 ng/g. Generally, strains containing *seb* or *sec* types were the most toxin-producing of this types of SE. However, strains containing only *egc* operon *se* genes seemed to produce the least enterotoxins. This was observed by several authors who concluded that *egc* enterotoxins were less expressed than the classical enterotoxins [27,49,50,51].

The five types, SEH, SEK, SEQ, SER, and SET, exhibited comparable concentration levels to the classical enterotoxins SEA, SED, and SEE. Type SEH was reported only once as being involved in a food poisoning outbreak due to the consumption of contaminated mashed potatoes. However, SEK, SEQ, SER, and SET were never detected in the context of food poisoning outbreak investigations due to the lack of methods targeting these enterotoxins [52]. Therefore, the present work showed that SEK, SEQ, SER, and SET should be studied for their toxicity and their possible involvement in SFPO.

## 3. Conclusions

This work highlights the usefulness of LC-MS methods for SEs multiplex analysis without the need for specific reagents, as is the case for immuno-enzymatic assays. A highly specific and sensitive LC-MS method was developed for the multiplex detection of 24 SE types in culture medium. The protocol was based on acid precipitation and enzymatic digestion followed by specific peptide detection. To enhance the accuracy of the method, a database of 93 signature peptides was optimized using the complete sequences of the 24 SEs, including their 162 variants.

The LC-MS method was successfully tested on 49 naturally contaminated BHI samples. The Hi3 procedure was adopted to overcome the absence of SE standard solutions for 19 SEs. The results showed the ability of CPS strains to simultaneously produce between 2 and 11 types of SEs. However, the concentrations expressed depended strongly on the type of SE. The lowest concentrations were obtained for enterotoxins SEG, SEI, SEM, SEN, SEO, and SElU expressed by *egc* operon (<1 ng/g). SEB and SEC were expressed at very high concentrations and reached the µg/g level.

The present study indicates that the newly described SEs should be taken into consideration during foodborne outbreak investigations, as CPS strains are able to simultaneously express several SEs, including these new SEs. LC-MS is a promising method for the multiplex detection of SEs and deserves further development to be applied to food.

## 4. Materials and Methods

### 4.1. Chemicals and Reagents

Ammonium bicarbonate (AB), brain heart infusion (BHI), dithiothreitol (DTT), iodoacetamide (IAA), analytical grade formic acid, hydrochloric acid (HCl), and trichloroacetic acid (TCA) were purchased from Sigma Chemical Co. (St. Louis, MO, USA), RapiGest SF from Waters Corporation (Milford, MA, USA), and ultrapure water from a Milli-Q plus purifier (Millipore, Bedford, MA, USA). Sequencing grade-modified trypsin was purchased from Promega (Fitchburg, WI, USA).

Highly purified, freeze-dried SEs were purchased from Toxin Technology, Sarasota, FL, USA (batch 120794A for SEA, 61499B1 for SEB, 113094C2 for SEC, 12802D for SED, and 70595E for SEE) and were rehydrated according to the manufacturer’s instructions to obtain stock solutions. In brief, 1 mL of osmosed water was added to 1 mg of SE powder to obtain a theoretical concentration equal to 1 mg/mL. The Uniprot (http://www.uniprot.org, accessed date 25 June 2021) sequence entries were P0A0L2, P01552, P34071, P20723, and P12993, respectively. Synthetic heavy peptides with labeled Lys [13C6; 15N2] (+8 Da) and Arg [13C6; 15N4] (+10 Da) were synthesized by Thermo scientific (Bremen, Germany) in Pepotec grade (Appendix A).

Safety Precaution: due to the high toxicity of SE, experiments were performed in accordance with safety rules for the handling of toxic substances and using personal protection equipment. SE-contaminated solutions and consumables were inactivated overnight with 2 M NaOH before elimination.

### 4.2. CPS Strains and Enterotoxin Sequences

A total of 49 CPS strains were used in this study: 6 reference strains obtained from the Institut Pasteur collection (France) and 43 CPS strains from the ANSES collection, including 40 strains isolated from different types of foods involved in staphylococcus food poisoning outbreaks. Three strains were isolated in the factory to check the contamination of the milk or cheese by CPS or SEs; this is called the own check. The origins of the strains are detailed in Appendix A.

The enterotoxin gene profiles were analyzed for their staphylococcal enterotoxin genes using the genomic tool NAuRA (https://github.com/afelten-Anses/NAuRA, accessed date 25 June 2021). Screening of the enterotoxins was performed using the gene sequence and the relative protein sequence of the previously described 27 SEs and the estimated parameters of Blast in Merda et al. Sequence alignment is detailed in Appendix A.

### 4.3. SE Expression in Culture Supernatant and Sample Preparation

Strains were incubated for 24 h at 37 °C on milk plate count agar. Colonies were then transferred to 40 mL of BHI at 37 °C for 18 h and centrifuged at 3000× *g* for 15 min to discard the bacterial pellet. The supernatant was filtered through a 0.2 µm filter before SE analysis, as in Lefebvre et al. 2021 [29,53]. A total of 1.2 mL of the filtered supernatant was centrifuged at 15,000× *g* for 10 min to discard residues. Then, 1 mL was recovered and mixed with 50 µL of TCA 100% for 5 min, and toxins were precipitated by centrifugation at 15,000× *g* for 10 min. The supernatant was removed, and the protein pellet was resuspended with 20 µL of RapiGest 0.05% in AB500 (Ammonium bicarbonate 500 mM). 10 µL of 20 mM DTT were added and heated for 15 min at 95 °C to induce protein denaturation. Then, 10 μL of 45 mM IAA was added and incubated for 30 min in the dark at room temperature. Digestion was performed with 1 µg trypsin for 3 h at 37 °C. After incubation, digestion was stopped with 5 μL of 1 M HCl and 10 µL of the mixture of labeled peptides was added to the samples. A total of 10 µL were sampled for LC-MS analysis.

### 4.4. Liquid Chromatography−Mass Spectrometry Analysis

LC-MS analysis was performed using an Ultimate 3000 chromatography system coupled to an Q-Exactive (Quadrupole-Orbitrap) mass spectrometer (Thermo Fisher Scientific, Courtaboeuf, France). Peptide separation was performed on an Aeris peptide XB-C18 reverse phase column (150 mm × 2.1 mm; 1.7 μm; 100 Å; Phenomex, Torrance, CA, USA) at 60 °C for 30 min. The mobile phases were prepared in water with 0.1% formic acid (A) and ACN with 0.1% formic acid (B). Following an isocratic step with gradient from 5 to 40% phase B in 20 min was carried out at a mobile phase flow rate of 0.5 mL/min, before equilibration for 5 min. Eluted peptides were introduced into the Q-Exactive instrument by an electrospray ionization (ESI) source. Data were acquired in positive ion mode, capillary voltage was set at 4 kV, and temperature at 320 °C.

DDA mode (Top 5) was performed with a resolution of 70,000 (Full MS) or 17,500 (ddMS2) at *m*/*z* 200 (FWHM), and normalized collision energy was fixed to 18%. The automatic gain control (AGC) target was constantly set at 1 × 10^6^, fill time was adjusted at 128 ms. For PRM mode, the isolation mass window was fixed at 1.2 Da. Collision energy was optimized in the HCD for each peptide, as indicated in Section 2.2 and shown in Appendix A. The AGC target was constantly set at 1 × 10^6^, fill time was adjusted at 128 ms, and Orbitrap resolution set at 35,000 at *m*/*z* 200 (FWHM) to reach at least 10 acquisition points per chromatographic peak. 

## Figures and Tables

**Figure 1 toxins-14-00249-f001:**
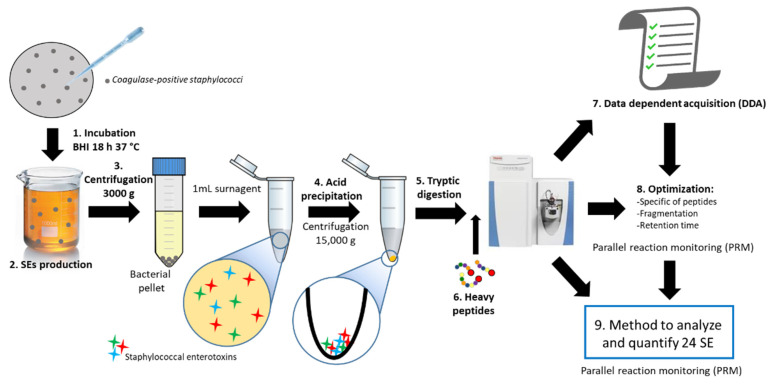
Workflow for detection and quantification of novel SEs: (1–2) production of SEs by CPS, (3) elimination of CPS by centrifugation, (4) concentration of SEs by acid precipitation, (5–6) tryptic digestion and addition of heavy peptides, (7) DDA analysis for peptide identification, (8) optimization of PRM for identified peptides by DDA, and (9) method for analyzing and quantifying SEs.

**Figure 2 toxins-14-00249-f002:**
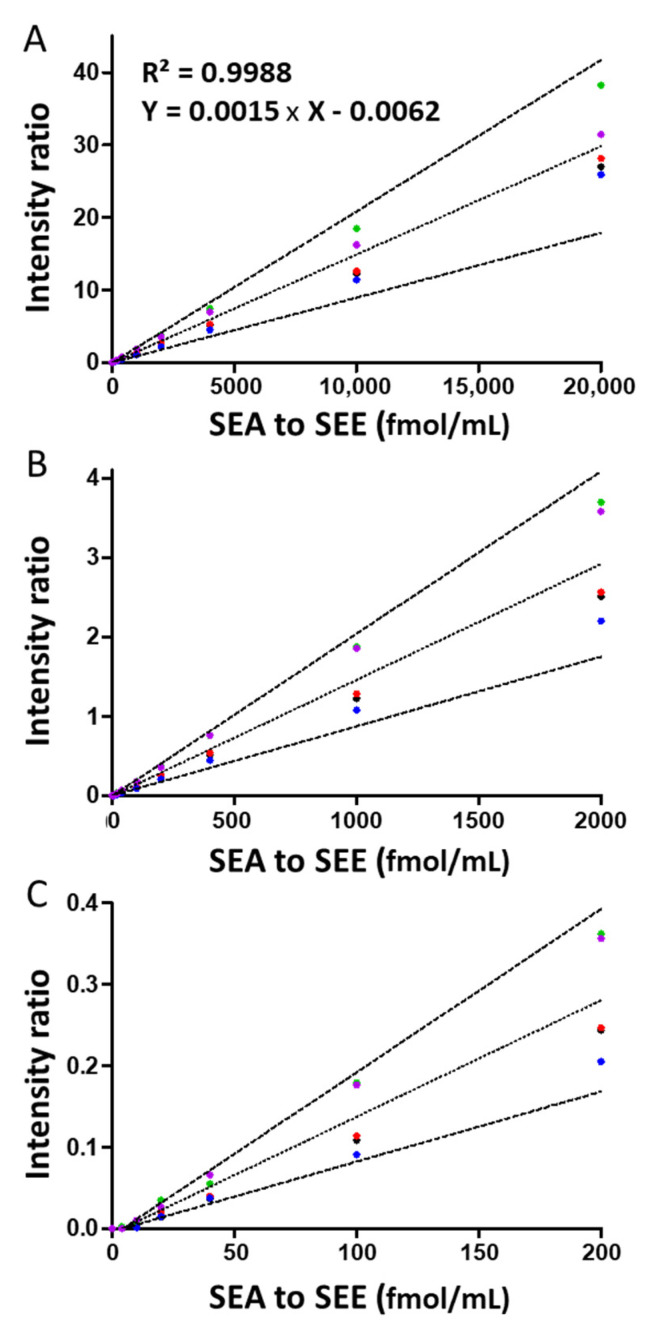
Calibration curve for SEA●, SEB●, SEC●, SED● and SEE● toxins. Mean of the five curves used for quantification (dotted line) and mean ± 40% (dashed line). (**A**) Range up to 20,000 fmol/mL. (**B**) Range up to 2000 fmol/mL. (**C**) Range up to 200 fmol/mL.

**Table 1 toxins-14-00249-t001:** Identification of genes coding for enterotoxins with the NAuRA tool in 49 strains. Each number corresponds to a variant coding for an enterotoxin.

Strain	sea	seb	sec	sed	see	seg	seh	sei	selj	sek	sel	sem	sen	seo	sep	seq	ser	ses	set	seu	selx	sey	selz	tsst1
05CEB51			1			1	1	1			1	1	1	19						17	17			
05CEB52		2				2		2				2	2	2						2	4	2	2	
05CEB53			3			3		3			5	3	3	3						5				
06CEB196	3			5					1	5						4	1				45			
06CEB83			8								6										4			2
08CEB393						3		4	3			15	3	3	2		1	3			9			
08CEB402																					36	7		
09CEB231	3			1					1								5				5			
09CEB303									2								2	1	1		5			
09CEB314					2																12			
10CEB282						3		4				6	3	3							9			
10CEB401	3		4							2	4					2					5			
11CEB110						7		4				4	7	6						5	34	6		
12CEB137			2								2										14			4
12CEB368		1								3						1					15	3		
12CEB496	3	3					3														16			
12CEB512						5		6				4	5	3						4	10	1		
13CEB193	2		2								2										7			2
13CEB437			3			3		3			10	3	3	3						6				
14A/FRI361			3	1		3		3	1		5	8	3	3			1			5				
14SBCL1004	3		10								4										5			
14SBCL770	1					4		10				5	4	4						1				3
15SBCL1151		1				3		4		7		4	3	3		5					9			
15SBCL1428						3		4				4	3	3						4	10	1		
15SBCL1438			7					11			1		6	5						9	22	4	1	5
15SBCL452	3					3		4	1			4	3	3			4				9			
16SBCL1028															3						1			
16SBCL1204						2		12				2	8	7						10	4	2	2	
16SBCL1368	4		1								8										5			1
16SBCL246						3		1				4		8						4	24			
16SBCL259					3																25			
16SBCL899	4					4		5				5	4	4						1				3
17SBCL25		3						13				9	9	9						11	28			
17SBCL330			5			3		3			12	8	3	3						5				6
17SBCL693	3		11				2			2	9					2					3			
18SBCL601							4														3			
18SBCL855				2		3		15	1			4	3	14	2		1				9			
19SBCL1059						3		4				18	3	3	2						9			
19SBCL591	3						2			9						2					3			
20SBCL08			1			7		4			7	4	7	6						5	3			
337E		4				3		4				4	7	21						5	41			
349E	3			5					4								1				5			
352E			13					8			1		6	5						9	22	4	1	5
356E						3		4				4	3	22						4	10	1		
42A/FRIS6	1	1								1						1					5			
43A/FRI137			1			1	1	1			1	1	1	1						7	17			
44A/FRI1230			3							4					4	1					5		3	
45A/FRI1151M				3					1								1				5			
46A/FRI326					1											3							5	
Number of variants	4	4	10	4	3	6	4	12	4	8	11	9	9	13	3	5	4	2	1	10	20	6	4	6

**Table 2 toxins-14-00249-t002:** Optimization and identification of best peptides for SEA: each peptide was identified by DDA analysis. Optimization for retention time (RT), mass/charge (*m*/*z*) ratio, normalized collision energy (NCE%), and fragment ion. WGS result for the representativeness of the peptide. Area ratio of peptide-to-peptide ratio QNTVPLETVK.

SEA Peptide	*m*/*z*	z	RT (min)	NCE%	Fragments ^(1)^	Detected ^(2)^	Not Detected ^(3)^	Shared Peptides ^(4)^	Peptide Ratio	Specific Peptide ^(5)^
GLIVFHTSTEPSVNYDLFGAQGQYSNTLLR	1109.8930	3+	14.9	18	y_9_^+^, y_10_^+^, y_12_^+^	2, 3	1, 4		0.67	Yes
NVTVQELDLQAR	693.3728	2+	9.5	21	y_9_^+^, y_8_^+^, y_10_^+^	1, 3, 4	2		1.27	Yes
QNTVPLETVK	564.8166	2+	7.3	18	y_6_^+^, y_6_^++^, y_8_^+^	1, 2, 3, 4			1.00	Yes
SELQGTALGNLK	615.8381	2+	8.5	21	y_8_^+^, y_9_^+^, y_10_^+^	3, 4	1, 2		0.18	
YNLYNSDVFDGK	717.8304	2+	10	18	y_8_^+^, y_9_^+^, y_10_^+^	1, 3, 4	2		0.77	Yes
GFFTDHSWYNDLLVDFDSK	769.3515	3+	15.9	18	y_5_^+^, y_6_^+^, y_7_^+^	3	1, 2, 4		0.18	
QIYYYNEK	560.7691	2+	6.0	18	y_4_^+^, y_5_^+^, y_6_^+^	1, 3, 4	2	SEE1, 2, 3	0.48	
SELQGAALGNLK	600.8328	2+	8.5	18	y_6_^+^, y_8_^+^, y_9_^+^	1	2, 3, 4		0.25	
GLIVFHTSTEPSVNYDLFGAQGQNSNTLLR	1093.5529	3+	14.4	18	y_9_^+^, y_10_^+^, y_12_^+^	1, 4	2, 3		0.76	
GFFTNHSWYNDLLVDFDSK	769.0235	3+	15.4	18	y_5_^+^, y_6_^+^, y_7_^+^	1, 4	2, 3		0.04	
EVTVQELDLQAR	700.8726	2+	9.8	18	y_6_^+^, y_7_^+^, y_8_^+^	2	1, 3, 4	SEE1, 2, 3	0.61	Yes
SELQGVALDNLR	657.8542	2+	10.1	18	y_5_^+^, y_6_^+^, y_8_^+^	2	1, 3, 4		0.86	

^(1)^ fragments selected for detection. ^(2)^ peptide was detected in variant (number of variant is available in Table 1). ^(3)^ peptide was not detected in variant (number of variant is available in Table 1). ^(4)^ peptide detected in another type of SEs. ^(5)^ five specific signature peptides were selected for the detection of type SEA, taking into account the four variants.

**Table 3 toxins-14-00249-t003:** Qualitative performance of the LC-MS method for the 24 enterotoxins.

	SEA	SEB	SEC	SED	SEE	SEG	SEH	SEI	SElJ	SEK	SEL	SEM	SEN	SEO	SEP	SEQ	SER	SES	SET	SElU	SElX	SEY	SElZ	TSST1	Total
Strains containing the *se* gene	14	7	17	6	3	23	6	26	9	8	16	24	25	26	5	9	9	2	1	20	42	10	6	9	323
Strains negative for targeted *se* gene	35	42	32	43	46	26	43	23	40	41	33	25	24	23	44	40	40	47	48	29	7	39	43	40	853
Corresponding enterotoxin detected	14	7	17	6	3	23	6	26	9	8	15	24	19	23	5	9	9	2	1	18	42	10	6	9	311
Positive deviation	0	0	0	0	0	0	0	0	0	0	0	0	0	0	0	0	0	0	0	0	0	0	0	0	12
Negative deviation	0	0	0	0	0	0	0	0	0	0	1	0	6	3	0	0	0	0	0	2	0	0	0	0	0
Specificity (%)	100	100	100	100	100	100	100	100	100	100	100	100	100	100	100	100	100	100	100	100	100	100	100	100	100
Sensitivity (%)	100	100	100	100	100	100	100	100	100	100	93.8	100	76.0	88.5	100	100	100	100	100	90.0	100	100	100	100	96.3
Accuracy (%)	100	100	100	100	100	100	100	100	100	100	98.0	100	87.7	93.9	100	100	100	100	100	95.9	100	100	100	100	99.0

**Table 4 toxins-14-00249-t004:** Bias (%) calculated for SEA, SEB, SEC, SED, and SEE at the 12 concentrations of the calibration curve, established according to the Hi3 procedure.

fMol	10	20	40	100	200	400	1000	2000	4000	10,000	20,000
SEA	−79.7	−49.4	−37.9	−22.6	−13.0	−9.0	−12.6	−10.5	−9.0	−12.6	−3.6
SEB	−53.6	−26.5	−28.6	−18.6	−11.8	−3.7	−8.3	−8.5	−6.4	−10.0	0.5
SEC	−43.3	24.6	−1.5	28.1	29.4	34.9	33.9	31.9	32.6	32.0	36.5
SED	−90.5	−47.6	−37.8	−35.2	−26.7	−20.1	−23.0	−21.5	−20.1	−18.7	−7.5
SEE	−26.7	−4.4	17.8	26.1	27.3	35.9	32.5	27.8	24.5	15.7	12.2

**Table 5 toxins-14-00249-t005:** SE concentrations measured in the 49 contaminated BHI samples using the LC-MS method, the signature peptides database and the Hi3 procedure. LOQ = 1 ng/mL; ^(1)^ blank box: *se* gene not detected by genomic analysis and SEs not detected by LC-MS; ^(2)^ <1: below LoQ; ^(3)^ ND: *se* gene detected by genomic analysis but corresponding SE not detected by LC-MS.

		Concentration of Staphylococcal Enterotoxins Type																							Total [SE] in the Sample
	SEA	SEB	SEC	SED	SEE	SEG	SEH	SEI	SElJ	SEK	SEL	SEM	SEN	SEO	SEP	SEQ	SER	SES	SET	SElU	SElX	SEY	SElZ	TSST1	
16SBCL1204	^(1)^					<1 ^(2)^		<1				<1	ND ^(3)^	<1						<1	1.4	2.0	<1		3.4
10CEB282						<1		<1				2.3	<1	<1							1.5				3.9
15SBCL1428						<1		<1				1.2	<1	<1						<1	<1	6.1			7.3
08CEB402																					5.1	2.8			7.9
08CEB393						<1		<1	<1			<1	<1	<1	<1		4.6	6.0			1.9				12.5
11CEB110						1.3		<1				3.8	<1	<1						1.6	<1	6.5			13.2
19SBCL1059						<1		<1				3.8	<1	<1	5.4						8.9				18.0
356E						1.1		<1				5.2	<1	<1						2.9	9.1	5.0			23.2
16SBCL1028															8.0						17.1				25.1
16SBCL246						1.1		<1				5.7		<1						2.7	17.0				26.6
337E		20.2				<1		<1				2.5	<1	<1						<1	7.2				29.9
15SBCL1438			2.2					<1			10.0		ND	ND						<1	<1	4.4	11.2	2.9	30.7
14SBCL770	13.7					<1		<1				<1	ND	<1						<1				28.7	42.4
12CEB512						<1		<1				4.2	<1	<1						1.5	32.0	5.0			42.8
18SBCL601							60.0														13.5				73.5
09CEB231	26.8			28.4					<1								17.1				4.5				76.8
16SBCL899	18.6					<1		<1				<1	ND	ND						<1				59.6	78.2
16SBCL259					44.5																46.8				91.3
09CEB314					23.0																70.1				93.1
45A				80.1					<1								15.2				2.1				97.4
12CEB137			65.2								10.1										1.7			38.2	115.1
18SBCL855				57.7		1.2		<1	<1			4.7	<1	<1	11.9		29.6				15.1				120.2
09CEB303									7.0								21.2	18.0	65.6		18.1				129.9
46A					106.1											27.7							36.9		170.7
15SBCL452	98.1					<1		<1	<1			4.3	<1	<1			30.5				38.4				171.3
349E	82.4			62.2					<1								31.4				4.6				180.7
352E			116.1					<1			6.4		ND	ND						ND	<1	3.7	5.6	73.0	204.8
06CEB196	39.9			67.4					<1	27.0						40.0	33.2				12.2				219.7
19SBCL591	56.5						62.6			32.4						69.7					25.9				247.1
43A			218.4			<1	121.9	<1			30.2	<1	<1	<1						ND	11.0				381.5
20SBCL08			393.7			1.9		4.6			18.9	7.2	1.3	<1						4.2	32.6				464.4
17SBCL693	76.5		157.4				32.4			49.2	10.3					112.4					31.1				469.3
16SBCL1368	28.2		167.2								11.1										45.7			386.9	639.2
44A			550.5							37.8					<1	61.4					9.7		5.9		665.4
05CEB52		710.8				2.6		1.3				2.3	<1	2.0						3.1	52.5	35.4	3.6		813.7
05CEB53			1731.3			<1		<1			3.4	1.8	<1	<1						1.0					1737.5
15SBCL1151		2434.4				3.6		6.9		41.3		12.6	1.4	2.7		95.8					62.1				2660.7
13CEB193	66.0		2592.1								6.9										<1			460.5	3125.5
12CEB368		3285.9								32.1						41.0					1.5	7.4			3367.9
17SBCL330			3371.7			<1		<1			7.2	2.7	<1	<1						1.4				25.1	3408.2
06CEB83			3323.9								79.5										6.5			129.0	3538.9
05CEB51			4341.8			<1	52.1	<1			12.5	<1	ND	<1						<1	33.8				4440.3
17SBCL25		5154.5						<1				1.0	<1	<1						<1	46.2				5201.7
13CEB437			5524.9			<1		<1			ND	2.6	<1	<1						1.5					5529.0
12CEB496	124.1	5651.8					45.9														29.5				5851.2
10CEB401	64.4		6238.2							20.5	13.5					44.5					9.5				6390.7
14A			7798.6	30.1		<1		<1	<1		7.9	2.0	<1	<1			47.1			<1					7885.7
14SBCL1004	94.1		8364.9								17.7										16.3				8493.0
42A	43.7	21,943.4								35.4						59.8					20.9				22,103.2
Median	60.5	3285.9	1731.3	60.0	44.5	<1	56.0	<1	<1	33.9	10.3	2.4	<1	<1	5.4	59.8	29.6	12.0	65.6	<1	11.6	5.0	5.8	59.6	
max	124.1	21,943.4	8364.9	80.1	106.1	3.6	121.9	6.9	7.0	49.2	79.5	12.6	1.4	2.7	11.9	112.4	47.1	18.0	65.6	4.2	70.1	35.4	36.9	460.5	
min	13.7	20.2	2.2	28.4	23.0	<1	32.4	<1	<1	20.5	3.4	<1	<1	<1	<1	27.7	4.6	6.0	65.6	<1	<1	2.0	<1	2.9	
SE number	14	7	17	6	3	23	6	26	9	8	16	24	25	26	5	9	9	2	1	20	42	10	6	9	

## Data Availability

The data presented in this study are available in this article, Appendix A and from the corresponding author.

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
