# Peer review of "Multiplex Detection of 24 Staphylococcal Enterotoxins in Culture Supernatant Using Liquid Chromatography Coupled to High-Resolution Mass Spectrometry"

_toxins, 2022, doi:10.3390/toxins14040249_

Round 1

Reviewer 1 Report

The authors describe a highly useful LC-MS method for the simultaneous detection and quantification of 24 different Staphylococcal enterotoxin in culture supernatant. The manuscript should be published in Toxins based on appropriately addressing the comments below.

Introduction

Line 34, 0.06 ng/g, is this a concentration in food which can already elicit certain symptoms in humans?

Line 65, Mass spectrometry is a useful alternative to immune-enzymatic methods

Line 72, either other emetic SEs, what about the six missing (5 + 8 + 6= 19)?

Line 87, Bacillus subtilis

Results and discussion
line 108 and Table S1, 4 of the 49 strains were isolated from non-food samples (column SE detected in Table S1 and text in line 108), but column sample origin in Table S1 gives 1x “rice” and 1x “coriander turkey”?

Lines 113 and, enterotoxin genes contain x variants

Line 160, can an example for further applications be given?

Figure 1, supernatant. According to the description in the text, the internal standards are only added after tryptic digestion, which is not really reflected in the picture.

Line 237, The overall accuracy

Line 275, Table 4

Line 292, Figure 2 shows calibration curves but not LOQs

Line 297, Table 4

Figure 2, A-C, SEA to SEE?

Table 5, abbreviated toxin types; IJ, IU, IX?

Conclusions

Line 363, line 309 explains 2-11 types of SEs

Materials and Methods

Line 383, osmosed water or ultrapure water?

Line 397, what is the meaning of “isolated from own checks”, can an example be given?

Line 424, isocratic step, how long?

References

Please indicate abbreviated journal names.

Supplementary Table S2, specific peptide instead of selected peptide?

Author Response

Reviewer 1

Comments and Suggestions for Authors

The authors describe a highly useful LC-MS method for the simultaneous detection and quantification of 24 different Staphylococcal enterotoxin in culture supernatant. The manuscript should be published in Toxins based on appropriately addressing the comments below.

Authors:

Thank you very much for reviewing our work, and for the relevant remarks and suggestion that we answered and we have taken into account for the revision.

Introduction

Line 34, 0.06 ng/g, is this a concentration in food which can already elicit certain symptoms in humans?

Authors:

This concentration is the BMD calculated for the type SEA in GUILLIER et al 2016. “The attack rates, i.e. the number of persons with symptoms divided the number of exposed people, according to the doses of SEA ingested were used for dose response modelling and Benchmark Dose (BMD) establishment. BMD may then be defined as the dose that induces effects in 10% of the exposed population (BMD10). Usually the BMD10 is not the value of reference. The lower 95%-confidence interval of the BMD10, also called the Benchmark Dose lower limit (BMDL10) is preferentially used”.

Phrase changed in the manuscript “The Benchmark Dose lower limit (BMD10) was estimated for type SEA to be 6.1 ng”.

Line 65, Mass spectrometry is a useful alternative to immune-enzymatic methods

Authors:

Done !

Line 72, either other emetic SEs, what about the six missing (5 + 8 + 6= 19)?

Authors:

Phrase added in the text: “No published work was found on the optimization of the detection of the other six emetic SEs (SEO, SEP, SER, SES, SET and SEY) by MS”

Line 87, Bacillus subtilis

Authors:

Done

Results and discussion
line 108 and Table S1, 4 of the 49 strains were isolated from non-food samples (column SE detected in Table S1 and text in line 108), but column sample origin in Table S1 gives 1x “rice” and 1x “coriander turkey”?

Authors:

The table S1 was revised. Origin of 05CEB52 and 05CEB53 was checked. The origin is human and the FPO was due to rice consumption.

Lines 113 and, enterotoxin genes contain x variants

Authors:

Done

Line 160, can an example for further applications be given?

Authors:

In this paragraph, “As a result, the established list of 93 specific signature peptides can be considered a reference database for further work on the detection of SEs using an LC-MS method, and could be used for further applications in food matrices.”

The example is the analysis of SE in food because our work was performed on SE produced in culture supernatant.

Figure 1, supernatant. According to the description in the text, the internal standards are only added after tryptic digestion, which is not really reflected in the picture.

Authors:

Figure revised

Line 237, The overall accuracy

Authors:

Done

Line 275, Table 4

Authors:

Done

Line 292, Figure 2 shows calibration curves but not LOQs

Authors:

This was a mistake, thank you for the observation. Figure 4 is deleted.

Line 297, Table 4

Authors:

Done

Figure 2, A-C, SEA to SEE?

Authors:

Thank you, done

Table 5, abbreviated toxin types; IJ, IU, IX?

Authors:

Done

Conclusions

Line 363, line 309 explains 2-11 types of SEs

Authors:

Done

Materials and Methods

Line 383, osmosed water or ultrapure water?

Authors:

Osmosed water was used for toxins reconstitution (obtained from Milli-Q® IQ Water Purification System)

Line 397, what is the meaning of “isolated from own checks”, can an example be given?

Authors:

Rephrased: three strains isolated in the factory for checking the contamination of the milk or cheese by CPS or SEs, this is called own check

Line 424, isocratic step, how long?

Authors:

20 min, text modified.

References

Please indicate abbreviated journal names.

Authors:

Thank you for the observation, this will be modified.

Supplementary Table S2, specific peptide instead of selected peptide?

Authors:

Done

Reviewer 2 Report

This manuscript is very well written and presents  successful detection and quantification of 24 SEs by C-HRMS, which will bring advances in the research of staphylococcal toxins. This reviewer comments minor points to be considered for revision as follows.

  1. There is non-uniformities in distinctive descriptions of enterotoxin and enterotoxin-like gene/protein. For example, in Table 5, only enterotoxin-like protein Z is shown as "lZ", but "l" is not added to enerotoxin J, U, and X. In addition, for enterotoxin genes, "l" (-like) is not added as seen in Table 1. As described in published literatures,  for genes, "l" should be added, as "selj", "selx", etc. These descriptions should be checked throughout the manuscript text. In addition, enterotoxin types shown in Table 3 and Table 5 are different. (e.g., "SEA" and "A")  This description should be identical in the same article.
  2. Page 8-9: Table1 should be corrected as Table 4.
  3. Line 116-118: Authors did not detect sev, selw, sel26, and sel27, and the reason was explained as just rarity of them. However, this may not be true and lead readers to misunderstanding. SElV encoded by selv was identified as recombinant protein between SEM and SEI (Thomas et al., Infection and Immunity, 74:4724, 2006). selw is not rare, but it is highly prevalent among S. aureus clinical isolates, and exhibit higher genetic diversity (Aung et al., Toxins, 12:347, 2020). sel26 and sel27 may be rare because of its presence in limited clones (ST27 and ST772 S. aureus, ST2250 S. argenteus) (Zhang et al., Int. J. Med. Microbiol. 2018, 308:438; Aung et al, Pathogens. 2021, 10:163). For these undetectable SEs, reasonable discussion should be made.

Author Response

Reviewer 2

Comments and Suggestions for Authors

This manuscript is very well written and presents  successful detection and quantification of 24 SEs by C-HRMS, which will bring advances in the research of staphylococcal toxins. This reviewer comments minor points to be considered for revision as follows.

Authors:

thank you very much for reviewing our work, and for the relevant remarks and suggestion.

The manuscript was revised taking into account all your suggestions.

  1. There is non-uniformities in distinctive descriptions of enterotoxin and enterotoxin-like gene/protein. For example, in Table 5, only enterotoxin-like protein Z is shown as "lZ", but "l" is not added to enerotoxin J, U, and X.

done

In addition, for enterotoxin genes, "l" (-like) is not added as seen in Table 1. As described in published literatures, for genes, "l" should be added, as "selj", "selx", etc. These descriptions should be checked throughout the manuscript text.

According to Omoe et al 2013 (doi: 10.1128/IAI.00550-13), the emetic potentials of SElK, SElL, SElM, SElN, SElO, SElP, and SElQ were assessed using a monkey-feeding assay. All these SEls that were tested induced emetic reactions in monkeys at a dose of 100 μg/kg.

  1. In addition, enterotoxin types shown in Table 3 and Table 5 are different. (e.g., "SEA" and "A")  This description should be identical in the same article.

The Table 5 is very large and we have a layout issue. It is not possible to add SE to all types. We will ask Toxins journal to help us!

  1. Page 8-9: Table1 should be corrected as Table 4.

done

  1. Line 116-118: Authors did not detect sev, selw, sel26, and sel27, and the reason was explained as just rarity of them. However, this may not be true and lead readers to misunderstanding. SElV encoded by selv was identified as recombinant protein between SEM and SEI (Thomas et al., Infection and Immunity, 74:4724, 2006). selw is not rare, but it is highly prevalent among S. aureus clinical isolates, and exhibit higher genetic diversity (Aung et al., Toxins, 12:347, 2020). sel26 and sel27 may be rare because of its presence in limited clones (ST27 and ST772 S. aureus, ST2250 S. argenteus) (Zhang et al., Int. J. Med. Microbiol. 2018, 308:438; Aung et al, Pathogens. 2021, 10:163). For these undetectable SEs, reasonable discussion should be made.

The phrase was revised in order to precise that this observation concern strains isolated from food samples.

However, in Merda et al, more than 400 S. aureus strains isolated from food samples were studied and confirm our observation. In addition, in our study, the selection of the 50 representative strains is issued from a collection of 630 S aureus strains and we confirm the rarity of presence of these genes in S aureus strains isolated from food samples.

Reviewer 3 Report

About this study:

The usefulness of LC-MS methods for multiplex analysis and in Staphylococcal food poisoning outbreaks caused by the ingestion of food contaminated with staphylococcal enterotoxins (SEs), and without the need for specific reagents, (as is the case for immuno-enzymatic assays) is presented in this nice work.

In this aim  highly specific and sensitive LC-MS method was developed for multiplex detection of SE types in culture medium. 

Hard points:

  • Introduction: well written with all needed info to link the research' topic.
  • M & M part is clear and fully replicable and supported by the obtained results.
  • R & D part, especially 2.2. Establishment of a signature peptides database for each enterotoxin type - excellent!
  • bibliography updated and well chosen.

What is improvable:

The LC-MS method was tested on 49 naturally contaminated BHI samples, and the Hi3 procedure was adopted to overcome the absence of SE standard solutions for 19 SEs - Provide here a phrase explaining more why was necessary.

Decision: one minor correction. 

Author Response

Reviewer 3

Comments and Suggestions for Authors

About this study:

The usefulness of LC-MS methods for multiplex analysis and in Staphylococcal food poisoning outbreaks caused by the ingestion of food contaminated with staphylococcal enterotoxins (SEs), and without the need for specific reagents, (as is the case for immuno-enzymatic assays) is presented in this nice work.

In this aim highly specific and sensitive LC-MS method was developed for multiplex detection of SE types in culture medium. 

Authors:

Thank you very much for reviewing our work and the positive feedback, remarks and suggestion. The end of introduction section was revised taking into account your suggestion

Hard points:

  • Introduction: well written with all needed info to link the research' topic.
  • M & M part is clear and fully replicable and supported by the obtained results.
  • R & D part, especially 2.2. Establishment of a signature peptides database for each enterotoxin type - excellent!
  • bibliography updated and well chosen.

What is improvable:

The LC-MS method was tested on 49 naturally contaminated BHI samples, and the Hi3 procedure was adopted to overcome the absence of SE standard solutions for 19 SEs - Provide here a phrase explaining more why was necessary.

Phrase add and of introduction « Finally, SE quantification is essential to study ES toxicity and to determine the emetic dose as Bench Marc Dose (BMD50). In this work, SE quantification was performed by the label-free Hi3 method in order to overcome the absence of SE standard solutions.